# Efficiency and Effectiveness of the European Parliament under the Ordinary Legislative Procedure

**Ani Matei** [1,*] **, Cristina Ciora** [2,*]**, Adrian Stelian Dumitru** [1] **and Reli Ceche** [1]

[1]   National University of Political Studies and Public Administration, 010643 Bucharest, Romania;
      adystelian@yahoo.com (A.S.D.); reli.ceche@oiposdru.edu.ro (R.C.)
[2]   Independent Researcher, 010164 Bucharest, Romania
*   Correspondence: amatei@snspa.ro (A.M.); cciora22@gmail.com (C.C.)

**Abstract:** In the aftermath of the 2019 European elections, the article tries to assess the efficiency and effectiveness of the European Parliament within the framework of the ordinary legislative procedure (co-decision). After defining and formulating the main indicators, the paper analyses the micro- and macro-performance of the European Parliament within the decision-making process from a quantitative-qualitative and a qualitative-quantitative perspective; highlighting the relativizing factors and the responsiveness of the European decision-making process to the Europeans' needs.

**Keywords:** European parliament; ordinary legislative procedure; co-decision; efficiency; effectiveness

## 1. Introductory Considerations

### 1.1. Purpose

In the context of the 2019 European elections, the 2017–2019 President of the European Parliament (EP), Mr. Antonio Tajani, put effectiveness at the very centre of his speech about Europe: 'We have to change Europe and make it more effective by answering citizens' concerns and building upon what we have already achieved' (European Parliament 2019a).

It is undoubtedly not without reason that effectiveness is approached as a primary way for answering the citizens' concerns. In May 2019, on the occasion of the European parliamentary elections for the ninth legislature, the European Parliament had to pass, in front of the European citizens, what we consider to be the ultimate test of effectiveness for the outcoming legislature And the high turnover of 50.62% (European Parliament 2019b) seems to confirm the perception of European citizens of a more effective and efficient European Parliament. In that context, the paper explores various ways of assessing the efficiency and effectiveness of the EP within the framework of the ordinary legislative procedure/the co-decision (OLP/COD) using quantitative and qualitative indicators appropriately identified as indicators of efficiency/effectiveness and efficiency/non-efficiency.

After theoretical considerations and a brief overview of the ordinary legislative procedure, we will formulate the qualitative and quantitative indicators and undertake a qualitative and quantitative analysis of the legislative production in the light of the EP.

### 1.2. General Considerations

'The decision-making process in the EU is a complex process determined by the plurality and the diversity of the Union (historical, traditional and cultural, economic and geographical)', Matei (2009, p. 24) states.

The co-decision procedure, which, since the entry into force of the Lisbon Treaty, has become the ordinary legislative procedure, is the common law procedure for decision-making at the European

level. Its complexity stems not only from the procedural toolbox, but also from the interests of the institutions involved in the decision-making process.

The main actors of the co-decision procedure are the three institutions of the 'institutional triangle', namely the European Commission, the Council of the European Union (CEU) and the European Parliament. The European Commission represents the general common interests of the Union and has the quasi-monopoly of the legislative initiative; the EP represents the interests of more than half a billion citizens and shares the legislative power with the CEU, which continues to represent the interests of the 28 Member States. The main secondary actors involved in the co-decision procedure are the advisory bodies of the European Union, namely the Committee of the Regions, representing the interests of the EU's regions, and the European Economic and Social Committee, representing the organised interests' groups.

Given the increasing complexity of the decision-making in the EU, many authors have been concerned about the efficiency and effectiveness of this process. Thus, Schulz and Konig (2000), with reference to 'institutional reform and decision-making efficiency in the European Union', argue that 'a common theme in the literature is that the efficiency of the EU decision-making process has deteriorated considerably as EU legislation activity has increased over the past two decades' (Schulz and Konig 2000, p. 653).

Neyer (2010) carries the debate deeper, addressing both the issue of efficiency and the effectiveness of EU decision-making processes. From Neyer's point of view, connections between for example national, supranational and intergovernmental entities call into question new aspects of the efficiency and effectiveness of the EU decision-making process. Neyer (2010, p. 19) tried to explain the 'unexpected efficiency and effectiveness in the European decision-making process'. It is extremely interesting that in 2004, the efficiency and effectiveness of the European decision-making process had been doomed to be 'unexpected', as though non-efficiency and lack of effectiveness would have been the general perception. Along with efficiency and effectiveness in the decision-making process, the author mentions within the decisional paradigm phrases such as 'the [EU] capacity to lead efficient and effective governance' and 'the degree of efficiency and effectiveness in European governance'.

In the context of the Eurocrisis, not only the efficiency and the effectiveness of EU processes but also the legitimacy of the processes in the context of democratic and institutional changes within the EU and the promotion of 'EU multi-tier governance' had been debated (European Parliament 2013). In addition, sectoral concerns such as 'environmental policies and co-decision' (Torres 2003) or thematic ones such as 'efficient and cost-effective interpretation in the European Parliament' (European Parliament 2013) are added to the overall concerns about efficiency and effectiveness.

At the European level, effectiveness represents one of the principles of good governance listed as such in the White Paper on European Governance launched in 2001: openness, participation, accountability, effectiveness and coherence. In this context, effectiveness is defined as follows: 'Policies must be effective and timely, delivering what is needed on the basis of clear objectives, an evaluation of future impact and, where available, of past experience. Effectiveness also depends on implementing EU policies in a proportionate manner and on taking decisions at the most appropriate level' (European Commission 2001, p. 7). One year earlier, in the White Paper on Reforming the Commission, effectiveness had been defined by the right action, depicting 'the extent to which objectives are reached and the relationship between the desired impact and the real impact of an activity'.

Effectiveness represents along with efficiency, one of the principles of good administration. Matei (2006, p. 192) defines effectiveness as 'the relationship between the outcome and the objective to be attained. This concept involves, on the one hand, the preliminary definition of an objective, and on the other hand, the measurement (or at least the estimation) of the outcome'. Furthermore, Matei (2006, p. 194) makes a distinction between the "macro" and "micro" effectiveness which are interdependent and indivisibly combined in a pertinent analysis. Thus, the author defines the two components depending on the importance of the objective to be attained as follows:

> "the "macro" effectiveness [targets] the impact of the action on the objectives of general interest to the company" (Matei 2006, p. 194)

"the "micro" effectiveness ( . . . ) concerns the effects of local operations with reference to the strategy of the company" (Matei 2006, p. 194).

The European Parliament representing the interests of 500 million citizens in the 28 EU Member States, must, at macro-level, ensure the representation of the interests of its citizens in the legislative projects, and, at micro-level, ensure the quality, quantity and speed of the decision-making process.

At the same time, approaching the effectiveness of legislation without considering its efficiency would be incomplete; the two concepts must be addressed at the same time, particularly because the quality of legislation and the effectiveness of the decision-making process are difficult to measure. As Dehousse (2011, p. 33) notes, 'efficiency requires [finding] a balance between quality, quantity and speed, and this is quite difficult'.

Efficiency represents "the ratio between the outcome and the employed means" (Matei 2006, p. 194). Bousta (2010, p. 175) defines efficiency as "the achievement of an objective with the least financial cost". However, we consider that this approach is limited, given the importance and complexity of the general public interest objectives.

The above is precisely the reason why the boundaries between efficiency and effectiveness of management often become porous in the sense provided by some authors.

## 2. Formulating the Assessment Indicators

### 2.1. Preliminary Considerations

'Everything that can be counted does not necessarily count; everything that counts cannot necessarily be counted' said Albert Einstein, quoted by (Dehousse 2011, p. 10).

Since a relevant analysis of the legislative production under the OLP/COD cannot be performed from an exclusively quantitative or qualitative perspective, we will proceed with two joint approaches that depend upon the pre-eminence of certain indicators: a quantitative-qualitative assessment and a qualitative-quantitative one. To return to one of our previous points, we emphasize Dehousse (2011) approach concerning the manager's difficult task of striking the right balance between quality, quantity and speed. 'The general objective of performance management is the continuous improvement of quality, efficiency and effectiveness by focusing on results and on the consequences of public services in relation to the internal processes' (Matei 2006, p. 197).

Before proceeding with the formulation of evaluation indicators, clarifications concerning the institution of the EP are necessary.

### 2.2. Institutional Considerations

Although the European Commission is the most interesting *sui generis* supranational institution, the EP is the only supranational institution whose members are democratically elected by direct universal suffrage, every five years. The last elections occurred in May 2019, and the next elections will occur in 2024. The Parliament represents the interests of European citizens, relying on democratic legitimacy. The EP currently has 750 members plus 1 for the President divided into 7 political groups, the first group being the European People's Party (European Parliament n.d.). The seats allocated to each state are assigned considering proportional regressive criteria (Bărbulescu 2015, p. 560).

The power of the EP has been strengthened over the years through its involvement in the European decision-making process, acquired in the context of treaty amendments. The process has made a fundamental contribution to the life and relevance of the EP in particular by increasing the EP legislative role from marginal to being a coequal of the CEU by its ability to prevent a measure being adopted without the approval of the CEU and the EP altogether (Craig and De Burca 2011, p. 23).

The 'birth certificate' of the COD/OLP, formerly known as the co-decision procedure, is the Maastricht Treaty, aiming at somehow addressing the 'democratic deficit' and at strengthening the democratic legitimacy of the EU. Under the Maastricht Treaty, the Parliament became co-legislator in 15 areas (European Parliament 2007a, p. 4). By the entry into force of the Treaty of Amsterdam,

the number of areas covered by the co-decision considerably increased to 38, providing 'a laboratory for institutional innovation and change' (Shackleton and Raunio 2003, p. 172). The Treaty of Nice extended the scope of the co-decision procedure to seven more areas (Infoeuropa 2007, p. 16).

After the entry into force of the Lisbon Treaty, the co-decision procedure, renamed the ordinary legislative procedure. has been regulated under art. 294 of the TFEU. The co-decision procedure comprises one, two or three readings, and a text cannot be adopted without the consent of the Parliament or of the Council altogether. After the entry into force of the Treaty of Amsterdam, an agreement can be obtained in one of the three readings, being no longer necessary to scroll through all the three stages. The third reading is preceded by a Conciliation Committee (European Union n.d.).

*2.3. Formulating the Quantitative Indicators*

With a view to the quantitative analysis of the evolution of the co-decision procedure, we can identify two types of main *quantitative indicators*:

(a)　number of procedures concluded in co-decision

- per legislature
- per reading
- per parliamentary committee

(b)　time needed to conclude a COD/OLP

Within the framework of the above-mentioned indicators, we can identify as indicators of efficiency the following:

- number of completed procedures
- number of procedures completed in the first or second reading
- decision-making time (length of procedure)

As non-efficiency indicators, we can identify the following:

- number of obsolete or withdrawn procedures
- number of procedures with a duration of more than 10 years
- number of rejected procedures

The number of acts adopted under COD/OLP can be an indicator of efficiency to assess the performance of the legislators. Conversely, in this case, there can be indicators of a qualitative nature, the purely technical dossiers being much easier to adopt than the dossiers with significant political implications. In this case, the quality of the adopted legislation can be more important than the number of adopted legislative acts.

The time necessary to complete a legislative procedure can be an indicator of efficiency or non-efficiency. A shorter decision time involves lower costs and, apparently, a higher efficiency. Conversely, the duration of a procedure can be the expression of the difficulty, complexity or sensitivity of a dossier.

To start with, a difference exists in approaching the life of a legislative procedure not only at the level of a certain involved institution but also within the same institution. Thus, if we examine the data provided by the Commission, Parliament or Council, we might notice small differences in terms of the number of files concluded under COD/OLP because the institutions have different interests affecting the counting of adopted files.

For example, at the level of the CEU, in which there is a half-year rotating Presidency (with the exception of the Foreign Affairs Council chaired by the EU High Representative and Vice-President of the European Commission for a period of 5 years), the number of concluded legislative procedures

can be an indicator to assess the performance of a given Presidency. Thus, even when a legislative procedure shall officially be deemed terminated after the act is signed by the Presidents of the Council and of the Parliament respectively and published in the Official Journal, the statistical data available on the website of the Council show as concluded legislative procedures those upon which an agreement had been reached.

Considering that, in a report published, in 2009, on the co-decision website of the European Commission constant reference is made to the data provided by the European Parliament, there is perhaps a greater convergence between the approaches of the European Commission and of the EP on COD/OLP.

Thus, on the former co-decision website of the European Commission, currently archived, the statistics considered the procedures from the legislative proposal until the signature of the legislative act.

At the level of the EP, there are three types of statistics, as follows:

- On the website of co-decisions and conciliations, the statistics present the procedures up to the date of signature.
- In the search engine of the Legislative Observatory (OEIL), those procedures whose acts have already been published or are awaiting publication in the Official Journal are considered complete.
- Various reports of the EP do not clearly specify when a COD/OLP is considered complete.

In conclusion, in this paper, although we approach the legislative production from the EP's perspective, primarily using the data supplied by the EP, we will note small or significant differences of quantitative order, depending upon the date used to consider a procedure complete, all of which we are perfectly aware.

Conversely, in our opinion, we appreciate that a legislative procedure should be considered formally concluded at the time of publication. Art. 297 TFEU stipulates that rules applicable to the legislative acts adopted under the ordinary legislative procedure shall be signed by the Presidents of the EP and CEU, respectively, as opposed to the legislative acts adopted in accordance with a special legislative procedure, which must be signed only by the President of the institution that adopted them. After the signature, the legislative acts shall be published in the Official Journal of the European Union, entering into force on the date provided for in the act or, in the absence thereof, on the twentieth day following its publication.

According to the data available in a study by the (European Commission 2009, p. 8), which targeted 900 acts adopted in co-decision from 1999 to 2009, the signature of the documents occurs on average 1.4 months after the adoption in the CUE (for files concluded at first reading). Alternatively, in the case of two readings, it occurs after the adoption in the plenary sessions of the EP. As reported in this study, in the case of 'early agreements' concluded early in the second reading, a subsequent adoption by the Council is no longer necessary, the reference point in this case being the adoption in the plenary sessions of the Parliament. According to the same study, following an analysis on 312 files, the average time between the signatures and the publication of an act is approximately 0.6 months (i.e., 17 days) (European Commission 2009, p. 8). The longest time from adoption until signature, recorded by the same study, was 3.5 months.

The stage (reading) in which the procedures are concluded might represent, at first sight, an indicator of efficiency or non-efficiency. Logically, without being a rule, the legislative procedures concluded in first reading are shorter than are those concluded on the second reading, which, in turn, are shorter than the procedures completed in third reading.

The sooner a procedure is completed (i.e., during the first or early second reading), the lower are the costs and, thus, the higher the efficiency of the decision-making process. The later a procedure is terminated (i.e., on the second or third reading), the higher are the costs for the adoption of that decision, thus possibly constituting an indicator of non-efficiency.

*2.4. Formulating the Qualitative Indicators*

*The qualitative analysis* of the legislation can be made according to the following:

- nature of the files
- quality of the legislation
- amendments tabled by the EP and taken over in the final text of the legislative acts

At a macro level, in a systemic evaluation, what counts in terms of quality is how the legislative and sectoral productions meet the needs of society, of the European citizens and of the economic and financial environment.

Unfortunately, as Albert Einstein, already quoted above, said, 'not everything that can be measured counts.' Furthermore, the qualitative indicators are those that are also often identified as indicators of effectiveness.

A first indicator in this respect can be the files adopted per parliamentary committee to assess to what extent the European decisions address the needs of European citizens and contribute to the setting up of a *Europe of concrete projects and results* for Europeans.

Conversely, by the nature of decisions, we can distinguish

(a)　*Primarily technical files*, which can be

- technical on the substance and of substantial nature (relating to the strictly technical areas de facto regulated at the level of expert groups)
- technical on the form and of formal nature (in the case of codifying, consolidating or repealing legislative acts in the framework of the Strategy for better regulation)

(b)　*Mostly political files*

Normally, the more sensitive a file is, the later it will be concluded in the ordinary legislative procedure (i.e., on the second or third reading), or it could even fail. Conversely, the more technical and uncontroversial a file is, the earlier it will be concluded (i.e., on the first or early second reading).

In the activity report of the Parliamentary Committee on the Environment, Public Health and Food Safety for the 2004–2009 legislature (European Parliament 2009a, p. 7), there is a distinction between *controversial* and *uncontroversial dossiers*.

As *uncontroversial* dossiers, the following categories are identified:

(a)　*Files of alignment to comitology* consisting of adapting the existing legislation to the 'regulatory procedure with scrutiny'
(b)　*Files* consisting *of strictly technical adaptations*, such as extending a certain transitional period (European Parliament 2009b, p. 7)

There is a certain hierarchy even in the classification of certain dossiers by their political nature. In the same report (European Parliament 2009b, pp. 8–9) there is, for example, a distinction between

(a)　*Purely political dossiers* such as the Climate package (consisting of files such as the Regulation on the reduction of $CO_2$ emissions produced by cars, the Emissions trading scheme, and the Regulation on cosmetics)
(b)　Sensitive *files from a political point of view* (such as LIFE+, Inspire, concluded on the third reading)
(c)　*Generally, more-sensitive files from a political point of view*, usually concluded on the second reading such as the Pesticides package or the two famous REACH pieces of legislation (the regulation and the directive)

However, the *purely political* nature of the files does not necessarily imply a longer legislative procedure, each rule being confirmed by its exceptions. Therefore, precisely the files referred to above

as 'purely political' (see Climate package) were concluded on the first reading because of 'political pressure' from 'inside or outside the European Parliament' (European Parliament 2009b, p. 9) and not on the second or third reading. This process is a relevant example of how politics can play the role of a procedural regulator, contributing to the early adoption of a decision.

Finally, the amendments specifically reflect the EP's intervention in the legislative project negotiated with the CEU. However, a strictly quantitative approach is flawed, considering that the quality of the tabled amendments, particularly of the adopted ones, can weigh more heavily in the decision-making economics. The EP's amendments, which can be directly found in the legislative texts published in the Official Journal of the European Union, directly affecting European citizens have, without doubt, the greatest political weight.

## 3. Assessment of the Legislative Work of the European Parliament in COD/OLP

*3.1. Quantitative-Qualitative Assessment of the EP's Legislative Activity*

A.　Legislative production per legislatures

Figure 1 presents an overview of the files concluded under the co-decision procedure, per readings, since 1993:

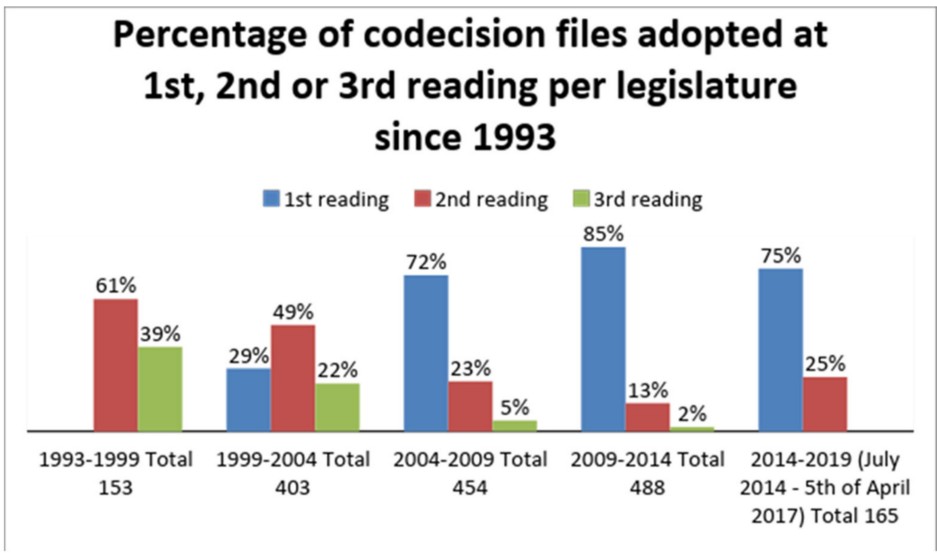

**Figure 1.** Percentage of co-decision adopted by 1st, 2nd or 3rd reading by legislature—all files included. Source: Various sources on the website of the European Parliament and official activity reports issued by the EP (European Parliament 2009b, p. 8; European Parliament 2014c, p. 8; European Parliament 2017).

According to the European Parliament (2007a) assessing, inter alia, the effect of the 2004 enlargement on the decision-making process, increasing the number of areas subject to this procedure following the treaties of Amsterdam and Nice, respectively, has naturally led to a substantial increase of acts adopted in co-decision in the 1999–2004 legislature. Thus, in the period 1999–2004, there had been adopted 403 legislative acts in co-decision. This number is 2.5 times greater than that of the co-decision files concluded in application of the provisions of the Treaty of Maastricht (1993–1999), namely 165 (European Parliament 2007a), although in Figure 1, 153 procedures appear concluded in the same legislature. The differences in these statistics had been previously explained.

The increase of the co-decision files adopted during the sixth parliamentary term (2004–2009) against the fifth from 403 to 454, almost tripling in comparison to the fourth term, can be a normal consequence of the entry under the co-decision procedure of certain areas, in 2005, after the entry into force of the Treaty of Nice, and of the high number of codifications and consolidations in the

framework of the European Commission's *Strategy for Better Regulation*. Furthermore, the fear of the old Member States concerning the functioning of the decision-making process after the 2004-accession of the ten new Member States also led to an increase in the number of co-decision files concluded before the end of the fifth legislature.

Conversely, we must note that the total number of files concluded under the COD/OLP in the seventh legislature (488) is only slightly greater than the total number of files concluded under COD/OLP during the sixth legislature (454) (European Parliament 2014a). Thus, the entry into force of the Treaty of Lisbon, particularly the almost double number of areas falling under the OLP, has not had any major effect on the legislative production. Given that the number of pieces of legislation concluded during the sixth legislature had been significantly increased by the legislative acts of codification and of alignment to comitology, we might assess the legislative output of the current legislature as being rather modest.

The high number of procedures completed in the first reading is a sign of a good understanding between the CEU and the EP. A contributing factor might be that, during the sixth legislature, the CEU and the EP had been politically closer due to the centre-right majority, which explains a tighter collaboration between the two institutions and a marginalization of the Commission, as proved by the adoption of the REACH Directive and of the Services Directive (Kurpas et al. 2008, p. 29). This increase can also be the sign of a 'higher familiarity' (European Parliament 2007b) between the components of 'the institutional triangle'.

Conversely, we can notice a significant increase in the percentage of files concluded on the first reading from 29% in the fifth legislature to 72% in the sixth and 85% in the seventh legislature, which represents an indicator of efficiency of the co-decision procedure as a whole. Faster decisions involve not only lower costs but, ex post, also a more rapid implementation of the policies codified in the legislative acts, thus addressing more quickly and timely the needs of European citizens. A longer procedure involves the risk of de-phasing the legislative acts from the developments of the economic and social environment and entails a certain deficit of efficiency. Similarly, in a logical manner, the percentage of the procedures adopted at second reading fell to less than half from 49% to 23% out of the total decisions adopted under co-decision.

The third and final stage of the procedure, known as 'conciliation', has become the exception and is limited to very difficult files. The percentage of cases adopted on third reading (after conciliation) decreased four times on average between the fifth and the sixth legislatures, from 22% to 5% reaching a rate of only 2% during the seventh legislature. Furthermore, at the mid of the eighth term, there had been no conciliation procedures for a period of 2.5 years. **(European Parliament 2017, p. 1)**. All of these indicators confirm the trend to constantly streamline the European decision-making process in general and the co-decision procedure in particular.

Figure 2 shows the number of procedures concluded in the last four of the legislatures, the fifth, sixth, seventh and mid-eighth.

First of all, one should note the distinction between the 2nd early reading and the 2nd reading, although they can sound similar. Considering that the EP and the Council can reach an informal agreement at any point, thus shortening the time of the procedure followed by the adoption of the file, since 2004 the co-decision has seen a new practice, namely the 'early second reading agreements' (European Parliament 2007b, p. 3).

One of the definitions of these agreements given by the EP is, "An 'early' second reading agreement is the product of successful negotiations between the Institutions after the Parliament has adopted its first reading position but before the Council has reached its common position ( . . . ). Although, formally speaking, procedures concluded in this way are concluded at [the] second reading stage, in reality, a political agreement has already been reached before Council completes its first reading" (European Parliament 2007b, pp. 11–12).

A study of the EP working group on institutional reform shows that, for 80% of the legislation adopted since July 2004, an agreement had been found informally, on first reading or at the beginning of the second

reading, the changes being directly integrated into the common position of the Council. It should be stressed out that the first reading has no time limit and is less formal (Kurpas et al. 2008, p. 30). Thus, the de facto number of dossiers concluded on first reading is actually greater than the number shown as closed de jure.

Figure 2 shows that almost 41% of the files completed during the sixth legislature, on second reading, have been the subject of an early agreement. Furthermore, during the 7th legislature, 60% of the procedures concluded on second reading rely on early second reading agreements. Early second reading agreements represent, along with dossiers concluded on first reading, efficiency indicators of the legislative process.

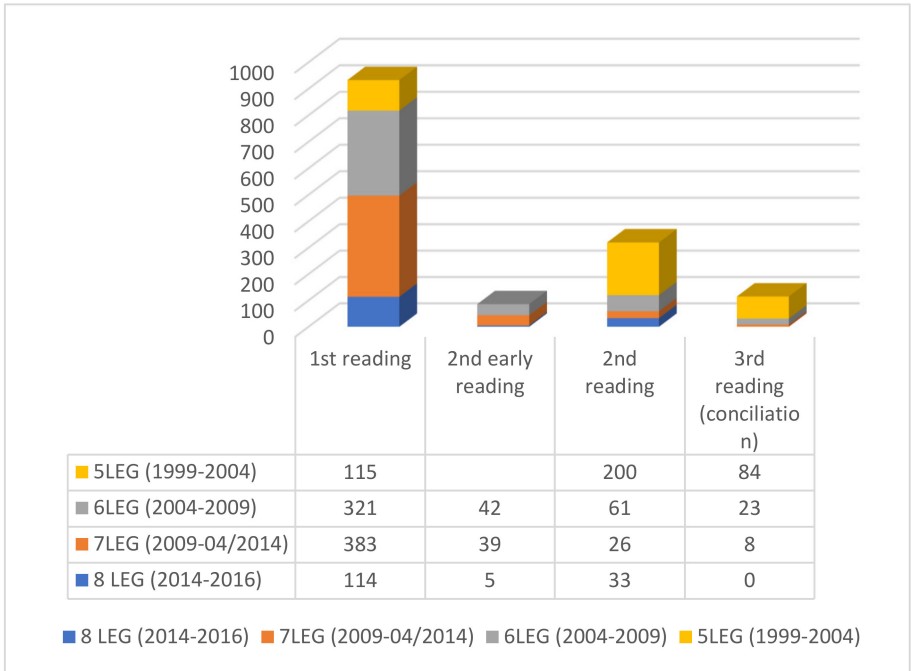

|  | 1st reading | 2nd early reading | 2nd reading | 3rd reading (conciliation) |
|---|---|---|---|---|
| 5LEG (1999-2004) | 115 |  | 200 | 84 |
| 6LEG (2004-2009) | 321 | 42 | 61 | 23 |
| 7LEG (2009-04/2014) | 383 | 39 | 26 | 8 |
| 8 LEG (2014-2016) | 114 | 5 | 33 | 0 |

■ 8 LEG (2014-2016)  ■ 7LEG (2009-04/2014)  ■ 6LEG (2004-2009)  ■ 5LEG (1999-2004)

**Figure 2.** Number of files concluded under co-decision (COD)/ordinary legislative procedure (OLP) per reading since 1999 (5th LEG) to 2016(mid-8th LEG). Legend: 5LEG: the fifth parliamentary term; 6LEG: the sixth parliamentary term; 7LEG: the seventh legislature; 8LEG: the on-going legislature; OLP: ordinary legislative procedure; COD: co-decision procedure. Source: Own creation based on data selected from various sources on the website of the European Parliament (For the figures relating to the sixth and the seventh legislature, we have used the statistics available on the Parliament website at http://www.europarl.europa.eu/code/about/statistics_en.htm last accessed on 19/04/2014. The figures on the fifth legislature were taken from the European Parliament (European Parliament 2004, p. 12). For the fifth legislature, there is no breakdown of statistics in the framework of the second reading between "early second reading" and "second reading. For the figures relating to the mid-8th legislature we used the data provided in the European Parliament (European Parliament 2017, p. 10).

Table 1, compiled by us, presents the state of play of all of the procedures per legislature

Because we have previously analyzed the evolution of completed procedures per legislature, we could furthermore identify as indicators of non-efficiency and non-effectiveness the lapsed or withdrawn procedures and, apparently, the rejected procedures.

The lapsed or withdrawn procedures represent indicators of non-efficiency and non-effectiveness in the sense that their withdrawal or lapse has financial effects; the money already invested in the elaboration and submission of proposals could not be recovered. Conversely, the objectives set therein can no longer be achieved timely and can no longer entail specific outcomes for European citizens. In 2014, one could identify a significant decrease in lapsed or withdrawn files from 68 in the fifth legislature to 26 in the sixth legislature, which outlines a continuous and gradual increase in the

efficiency of COD/OLP. However, when measuring the same data in 2019, using the same research engine, the decrease would appear less significant from the fifth (76) to the sixth (56) legislature. Furthermore, the decrease of withdrawn or lapsed files surprisingly turned into an increase from the sixth legislature (56) to the seventh one (67), That increase came as a natural result of the "EU Regulatory Fitness" policy of the European Commission launched in 2012 (European Commission 2012). On the other hand, when looking at the data provided by OEIL for the eighth legislature, one could note only 7 lapsed or withdrawn procedure, testifying for an obvious better efficiency and effectiveness of the co-decision procedure. Still, when consulting the data provided by the same European Parliament in an activity report from 2017 (European Parliament 2017, p. 13), reference is made to 41 withdrawn legislative proposals at the mid-term of the eighth legislature, all as a result of the Junker Commission policy on withdrawing legislative proposals that 'become obsolete due to scientific or technical advances or if they are no longer in line with new policy objectives' (Idem). A partial explanation can be found in the same Activity Report of the EP, outlining that '19 of these withdrawals concerned pending legislative proposals that the 8th Parliament had, at the start of its term, decided should be continued'. Thus, we may assume that some withdrawn proposals are to be found under the 'on-going procedures' label in the table above. Still, it is obvious that the huge difference in figures would need further clarifications.

**Table 1.** State of play of COD/OLPs during the last four legislatures (1999–2019).

| Type of Procedures | The Fifth Legislature (1999–2004) Measured in 2014 and 2019 | | The Sixth Legislature (2004–2009)Measured in 2014 and 2019 | | The Seventh Legislature (2009–2014) | The Eighth Legislature (2014–2019) |
|---|---|---|---|---|---|---|
| Completed procedures | 495 | 495 | 527 | 527 | 558 | 383 |
| Lapsed or withdrawn procedures | 68 | 71 | 26 | 56 | 67 | 7 |
| On-going procedures | 3 | 0 | 31 | 1 | 28 | 140 |
| Rejected procedures | 3 | 3 | 2 | 2 | - | |
| Total procedures | 569 | 569 | 586 | 586 | 653 | 530 |

Source: Own creation using the data provided through OEIL by the European Parliament (2014b) for the fifth and six legislatures (European Parliament 2014b) and the European Parliament (2019c) for the fifth, sixth, seventh and eighth legislatures respectively (European Parliament 2019c).

In that context, it is worth noting that, for the first time, recent case law (C-409/13 Commission v Council, judgment of 14 April 2015, EU:C:2015:217. The case concerned a proposal for a regulation laying down general provisions for macro-financial assistance to third countries (2011/0176 COD) of the Court of Justice has confirmed the Commission's right to withdraw its legislative proposals under specific conditions. Thus, in its judgment of 14 April 2015 in case C-409/1333, the Court of Justice for the first time analysed, and thereby clarified the scope of the Commission's right to withdraw its legislative proposals, pursuant to Article 293 (2) TFEU (European Parliament 2017, p. 13).

The Court of Justice recalled that the Commission has the right to withdraw a proposal at any time during the legislative procedure as long as the Council has not acted. However, the Court specified, "This was not a 'right of veto', and was necessarily circumscribed by the prerogatives of the other institutions. Furthermore, and in any case, a withdrawal by the Commission had to be appropriately justified to the co-legislators and, if necessary, supported by cogent evidence or arguments" (European Parliament 2017, pp. 13–14).

Concerning the rejected procedures, they appear to represent rather the exception to the rule, the search engine of the OEIL (European Parliament 2014b) displaying only 3 such procedures during

the fifth legislature and 2 during the sixth legislature. Although the rejected procedures can apparently be interpreted as an indicator of non-efficiency and non-effectiveness, the very low number thereof appears to indicate rather an efficient and effective decision-making system. Conversely, the complexity of the files and the public interests of European citizens cannot be quantified. Thus, if the entry into force of certain pieces of legislation would have had adverse effects on European citizens represented within the EP, then the rejection of those procedures could be beneficial and, paradoxically, even represent an indicator of effectiveness in the sense that the role of the EP to defend the general interests of Europeans had been successfully completed (Ciora 2013, p. 210).

For example, one of the two procedures rejected at the end of the sixth legislature refers to the Working Time Directive, when the EP could not agree, inter alia, on the counting of working time per contract and not per person. The rejected procedure directly concerned the protection of European workers, at the same time being a dossier of major political importance, particularly in the context of the current economic and social crisis (Ciora 2013, p. 210).

B.   Duration of COD/OLP

According to Peter F. Drucker, 'the first step towards efficiency is a procedure—keeping track of how you spend your time. ( ... ) If you keep it with some continuity, this record keeping will lead the man to the next step for greater efficiency' (Drucker 1994, p. 222).

A famous quote says that 'time is money'. Indeed, an extension of the decision-making time involves additional costs. However, as Einstein said, 'Everything that counts cannot necessarily be counted'. Thus, it is very difficult to set an optimal deadline within which a decision should be made, because everything depends upon the complexity of the legislative act. Therefore, it is difficult to identify the limit beyond which the passing of time entails additional unjustified costs.

Figure 3 illustrates the evolution of the average duration in months of a COD/OLP from the submission of the proposal during the last four legislatures until 2016.

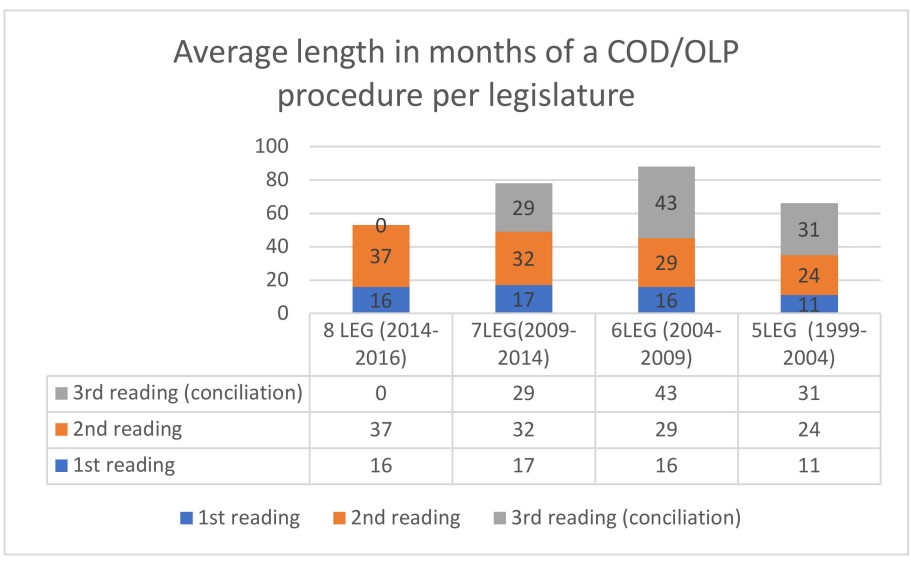

**Figure 3.** The average time, in months, of a COD/OLP since 1999 (5th LEG to 2016 (mid-8th LEG). Source: Own creation using the data provided, in 2017, by European Parliament (European Parliament 2017, p. 12).

As a general trend, we can observe, from one legislature to another, a constant increase in the duration of the procedures concluded during the first and second readings. Nevertheless, concerning the files concluded at the 3rd reading, following a conciliation procedure, there is a consistent increase of the decisional time in the sixth legislature compared with the fifth legislature, which is followed by a consistent decrease during the seventh legislature. Thus, in temporal terms, at least during the first

two readings, the legislatures apparently appear to have become less efficient from one to another, longer procedures entailing higher costs.

As regarding the duration of the procedures concluded at second reading, there can be noticed a constant increase from one legislature to another. Compared with the 7th Parliament, for instance, the slight increase for second reading files we can observe during the 8th LEG (and thus also for the total average length of all concluded co-decision procedures) can most likely be explained by the fact that almost all of these files were 'carried over' from the previous term, which inevitably implied delays during the transition from one Parliament to the next (European Parliament 2017, p. 12).

However, this interpretation must be relativized by the number of procedures completed in each reading. Thus, although the duration of a procedure which was completed on third reading is significantly greater than the duration of a procedure completed on first reading, the number of procedures completed on third reading is very small compared to the number of procedures completed on first reading.

In this context, in Figure 4, we tried to calculated the weighted average duration of a procedure completed during three legislatures.

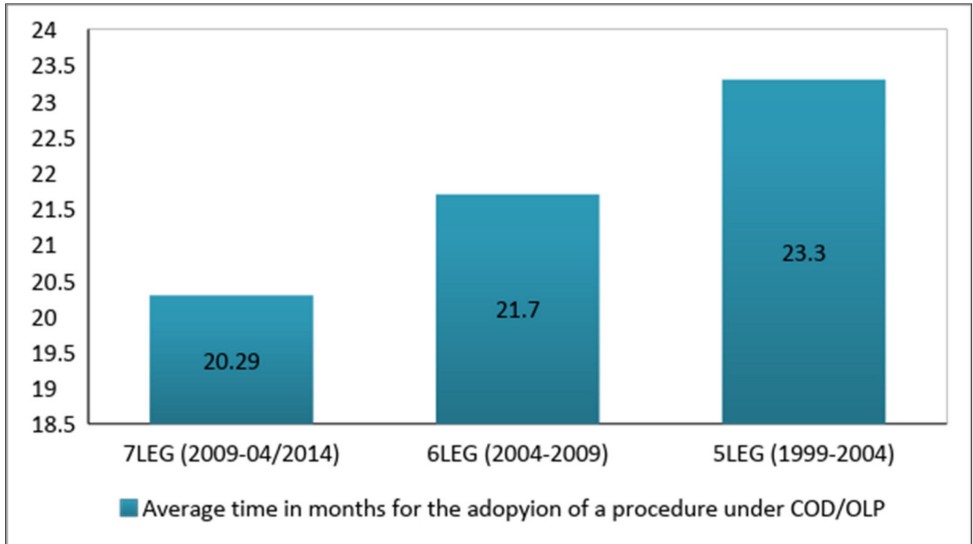

**Figure 4.** Average duration in months/per procedure of the files concluded under COD/OLP during three legislatures of the EP. Source: Corduban (2015, p. 128).

Surprisingly, Figure 4 shows a constant decrease of the weighted decisional time from one legislature to another, confirming the trend of continuous and progressive efficiency of the European decision-making process, in spite of the entry into force on the 1 December 2009 of the Lisbon Treaty and of the enlargements of 2004 and 2007 respectively.

Moreover, from a study performed by the European Commission, Table 2 shows huge differences in the duration of procedures concluded within the same reading:

**Table 2.** Average duration in months of the procedure from the Commission's proposal until the signature of the act (including the acts signed up until 13 May 2009).

| Agreement on | 1999–2004 Legislature | 2004–2009 Legislature | The Shortest and Longest Procedure (2004–2009) |
|---|---|---|---|
| First Reading | 13.8 | 15.2 | 1.8/47.9 |
| Second reading | 25.1 | 31.3 | 11.9/108.1 |
| Conciliation | 31.9 | 43.7 | 28.8/159.4 |

Source: European Commission (2009, p. 7).

Thus, during the 6th legislature (2004–2009), if the average duration of a COD procedure completed on first reading was 15.2 months, the shortest procedure lasted 1.8 months, whereas the longest one lasted 47.9 months. Undoubtedly, the shortest procedures refer to codifications, repeals or alignment to comitology files, whereas the longer ones relate to complex and sensitive dossiers from a political point of view.

In the case of procedures concluded on second reading, after an average period of 31.3 months, the shortest procedure ended after 11.9 months, whereas the longest procedure took approximately 108, 1 months.

The longest procedure of the sixth legislature took 159.4 months, more than 13 years, being completed on third reading after conciliation. The delay undoubtedly comes from the first reading, on which there is no temporal limit. At the opposite pole, the shortest procedure completed on third reading lasted 28.8 months, the average duration of the procedures concluded at this stage being 43.7 months.

Conversely, the data provided in Table 1—State of play of COD/OLPs during the last four legislatures showed that in 2014 there had been 3 on-going procedures from the 5th legislature, one opened in 2000 and two in 2003, whose duration of approximately 14/11 years was a clear indicator of non-efficiency and non-effectiveness considering that over time, the proposals had most likely been outdated by socio-economic developments in the European Union. Those files are three of those nicknamed in the EP's jargon 'sleeping beauties' and which outline the risks stemming from the lack of any deadlines during the first reading.

The three procedures concerned

(1)    Environment: public's right of access to information, right to participate in decision-making and right to justice (Aarhus Convention-2003/0246 (COD))
(2)    Travel services: indirect taxation (VAT), administrative cooperation-2003/0057 (COD)
(3)    Oil pollution: fund for damage compensation in European waters, Erika II package-2000/0326 (COD)

From a qualitative point of view, one can observe, without a doubt not incidentally, that two of the three 'sleeping beauties' fell within the competence of the Parliamentary Committee on Environment, Public Health and Food safety (ENVI), the dean of age celebrating at least its 14th anniversary as, in 2019, all 3 on-going procedures seem to be found under the withdrawn or elapsed procedures of the 5th legislature.

*3.2. Assessing the Qualitative-Quantitative Legislative Activity of the European Parliament*

A.    The nature of the adopted legislative dossiers

The qualitative nature of the legislation adopted at the European level might be an indicator of effectiveness in the sense of the useful and relevant representation of European citizens' interests.

The nature of the adopted acts might determine the length of a procedure. Thus, in the case of uncontroversial files, the procedure will be shorter, whereas the politically sensitive dossiers will most likely imply complex negotiations between the European Parliament and the Council.

Thus, during the sixth legislature, 106 of 447 files were of an uncontroversial nature, as follows:

(a)    46 of the 83 dossiers concluded by Committee on Legal Affairs (JURI) were legal codifications concluded on first reading (European Parliament 2009b, pp. 8, 11).
(b)    54 dossiers concerned the comitology alignment (to the new regulatory procedure with scrutiny).
(c)    6 were repealing dossiers (European Parliament 2009b, p. 9).

In this context, if we excluded from the total procedures concluded during the sixth legislature the uncontroversial files, we would count 341 new dossiers of substance concluded during the sixth legislature.

On the other hand, we should not oversee the format in which certain files are presented. As the European Commission has preferred to present various proposals in packages, it may happen that non-controversial pieces of legislation may appear more difficult to pass only due to their positioning in certain legislative packages encapsulating at least one sensitive file impacting upon the adoption time of the package as a whole.

B.   Distribution of COD/OLP files per Parliamentary Committee

The legislative dossiers concluded by each parliamentary committee in part represent a sectoral barometer of effective exercise of representative democracy, particularly given that the European Parliament is the one that 'emphasizes the role of non-economic factors in the co-legislating process' (Matei and Dogaru 2012, p. 120). The MEPs must 'resonate with the society', that is, with the interests of the European citizens whom they represent in the legislative process. The concept of 'resonator' has been used by Matei and Matei (2010, p. 116) in a broad sense, the public administration being considered 'a resonator of the society ensuring the interface with the citizens using its services'.

Each parliamentary committee shall ensure the representation of the interests of European citizens in the legislative acts and the internalization of certain interests through the tabled amendments. Essentially, the legislative acts adopted by area can indicate the extent to which the '*Europe of concrete results*' for Europeans is configured through the adopted legislative acts.

Over time, each parliamentary committee has developed its own culture, the oldest committees gaining an institutional memory affecting the very present. 'The political culture of the committee most likely plays a role; in the past, ENVI has proved reluctant to abandon positions of principle and to compromise early in the legislative procedure' (European Parliament 2007b, p. 4). Thus, ENVI has gained a reputation of a committee 'greener than the Parliament as a whole' (European Parliament 2007b, p. 4). However, the differences of position between the competent committee and EP's plenary can lead the procedures in the conciliation phase, which would involve a longer duration of the procedure, i.e., higher costs.

The culture of ENVI has been determined by its supremacy in number of handled files from one legislature to another, which entailed a gain in procedural experience superior to other committees. This is due to the following:

(1)   on the one hand, to the disastrous consequences for public opinion from the crises that have marked food safety and public health (such as mad cow disease and the avian flu), causing intense legislative activity in these areas, and

(2)   on the other hand, particularly during the fifth and the sixth legislature legislatures, to the emergence of themes related to environment and energy and to the need to promote a '*Europe of concrete projects*.'

According to the data provided by the European Parliament (2012, p. 3) at the mid-term of the seventh legislature, as in the previous legislature, ENVI occupied the first place on managed files, with a rate of 15.2%. ENVI was followed by the Committee on Economic and Monetary Affairs (ECON), with 13% of the files, and the Committee on Transport and Tourism (TRAN), with 11.9% of the files. Unlike the previous term, the ECON Committee changed places with the TRAN Committee, which is understandable because of the measures required in the context of the economic and financial crisis (Ciora 2013, pp. 211–12).

During the sixth term (2004–2009), the ENVI Committee clearly led with 20% of the concluded files, followed by the Committee on Legal Affairs (JURI) with a percentage of 18.3%, the TRAN committee with 11.4% and the ECON committee with 8.8% (European Parliament 2009b, p. 8). Given that 46 of the 83 (European Parliament 2009b, pp. 8, 11) files concluded by the JURI committee had been codifications, we can consider that the second and third places of the sectoral legislative podium during the sixth legislative term had been occupied by the TRAN and ECON committees with 52 and 40 files, respectively (European Parliament 2009b, p. 11). Finally, during the fifth legislature (1999–2004),

the ENVI Committee took first place with 117 files, followed by the TRAN Committee (72 files) and the JURI Committee (48 files) (European Parliament 2004, p. 11).

C.   The share of amendments in the adopted legislative acts

The amendments represent a primary, specific tool of influencing the legislative production by codifying the interests of the European citizens represented by MEPs. The number of amendments tabled by the EP and in particular of those taken over in the final published legislative texts represents an indicator of the effectiveness of MEPs' negotiations within the framework of the legislative process.

According to the French MEP Lamassoure (2008), 40% of the texts published in the Official Journal of the EU emanate from the European Parliament's amendments. However, in France, only 4% of the adopted legislative texts emanate from the General Assembly. Furthermore, in the United Kingdom, only 0.4% of passed laws emanate directly from House of Commons amendments. The percentage mentioned at the European level appears huge, given that those statistics referred to a period going back more than one year before the entry into force of the Lisbon Treaty.

In this framework, in terms of the amendments integrated into the final text of the laws, the European Parliament would appear10 times more effective than the French General Assembly and 100 times more effective than the UK House of Commons.

The shift of the legal paradigm seems to have started before the entry into force of the Lisbon Treaty. Thus, between 1994 and 2004, 809 (approximately 60%) of the proposed amendments (1344) during 86 conciliation procedures had been accepted (European Parliament 2004, p. 13), constituting a very high percentage and a relevant indicator of effectiveness for the EP.

## 4. Final Remarks

In a Europe seriously marked by a former economic and social crisis and ever challenged by topics such as Brexit, immigration and climate change finding the optimal balance between the amount of legislation, its quality, the speed of adoption and particularly its inherent costs represents a serious endeavour for any manager, particularly at the level of the European Parliament representing European citizens directly affected by the measures taken at the European level.

The co-decision procedure/OLP has constantly evolved towards growing efficiency by increasing the number of adopted files, along with decreasing the duration of procedures from one legislature to another. At the same time, the percentage of the files concluded at first reading constantly increased, whereas the number of conciliations and the number of rejected procedures constantly decreased. If at the mid of the eighth legislature, there had been recorded no conciliation procedure for the first time during the first half of a legislature, the seventh and the eighth legislatures had both seen no rejected procedures.

The effectiveness of co-decision had been proved through the sectorial breakdown of pieces of legislation echoing the main needs of European citizens, the more active parliamentary committees turning out to be the ones addressing precisely matters of highest concern for European citizens such as environmental, economic or social ones. Furthermore, the shorter time spent for adopting the procedures guaranteed that the concerns of the society are timely addressed. The management of legislation by thematic packages and the constant assessment of its 'fitness for purpose' by withdrawing the obsolete files has been steering the whole decision-making mechanism towards the path of a systemic effectiveness.

However, considering that twelve four-day plenary sessions per year in Strasbourg cost European citizens approximately 180 million euros per year and the environment approximately 19,000 tonnes of $CO_2$ (Singleseat n.d.), any analysis relating to the efficiency and effectiveness of the European Parliament in the decision-making process might appear somehow nonsensical.

According to Standard Eurobarometer 88 performed in November 2017, the confidence of European citizens in the European Union 'has reached a modest threshold' of 41%, although greater than their confidence in national parliaments (35%) and national governments (36%) (European Commission 2017,

p. 12). Nevertheless, since autumn 2015, trust in the EU has 'increased almost continuously' and remains at its second highest level in several years. On the other hand, 'distrust has risen slightly' since spring 2017: +1% for the EU, but −48% 'tend not to trust'.Compared with previous years, the main concerns of European citizens are now immigration and terrorism (39%), with the economic situation being in third place (17%), the situation of public finances in the Member States in fourth (16%) and unemployment in fifth (13%) (European Commission 2017, p. 4).

On the other hand, the unexpected high turnover at the 2019 European elections seems to highlight a different trend towards an ever-greater confidence of European citizens in the European Union, particularly challenging the European Parliament to find new ways of continuously improving its effectiveness and efficiency by timely addressing the most important concerns of the people.

**Author Contributions:** This article had been elaborated under the direct coordination and supervision of A.M. Data curation, A.S.D.; Investigation, R.C. The main part of this article comes from a subchapter of a PhD Thesis in Administrative Studies on the Managerial Approach of the European Ordinary Legislative Procedure defended by Cristina Corduban (Ciora) in 2015 at the NSPSPA of Bucharest. The elaboration of that subchapter had been directly supervised by Late Lucica Matei. The PhD Candidates elaborated Figure 1 and added some paragraphs all over the article, mainly referring to data analysed during 2017. C.C. updated the article with data referring to 2019.

**Funding:** This research received no external funding.

**Acknowledgments:** To Lucica Matei, as a tribute for her dedication in introducing and furthering new research on European Integration within higher education.

**Conflicts of Interest:** The authors declare no conflict of interest.

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
