# Peer review of "Efficiency and Effectiveness of the European Parliament under the Ordinary Legislative Procedure"

_admsci, doi:10.3390/admsci9030070_

Round 1

Reviewer 1 Report

From a theoretical perspective, I found very interesting the analysys of EU Parliament legislative procedure based on quantitative and qualitative criteria and parameters. I believe that this research could be a useful base for a deeper study on the crisis of mechanisms of legitimacy of EU polity.

Author Response

Thanks for your appreciation. We have completed completed in the sense of using them for further studies. I also reviewed the English translation.

we also tried to improve the quality of the paper by replacing less clearer charts with better and complete ones and by updating a table.

As regarding the language issues, minor corrections have been made.

Reviewer 2 Report

The manuscript could be improved in a few ways:

First, the authors should define efficiency and effectiveness sooner in the introduction and build from those definitions when discussing the figures later in their work.

Second, when considering the passage of files, it is important to remember that this items do not occur independently and may depend on the relative ease or difficulty of acting on other issues at the same time. Some discussion of this phenomenon is warranted.

Third, the conclusion could be improved by providing a succinct but comprehensive summary of the results from all of the tables and figures.

Finally, the discussion of Brexit does not seem to fit the overall context of the paper and it is abandoned after the introduction of the manuscript. I recommend either (1) bringing the Brexit discussion back in to add context to the findings later in the manuscript or (2) eliminating the discussion of Brexit completely.

Author Response

we also tried to improve the quality of the paper by replacing less clearer charts with better and complete ones and by updating a table.

As regarding the language issues, minor corrections have been made.